# Characterization of the Complete Mitochondrial Genome of a Whipworm *Trichuris skrjabini* (Nematoda: Trichuridae)

**DOI:** 10.3390/genes10060438

**Published:** 2019-06-09

**Authors:** Awais Ali Ahmad, Muhammad Abu Bakr Shabbir, Yang Xin, Muhammad Ikram, Mian Abdul Hafeez, Chunqun Wang, Ting Zhang, Caixian Zhou, Xingrun Yan, Mubashar Hassan, Min Hu

**Affiliations:** 1State Key Laboratory of Agricultural Microbiology, Key Laboratory for the Development of Veterinary Products, Ministry of Agriculture, College of Veterinary Medicine, Huazhong Agricultural University, Wuhan 430070, Hubei, China; awais@webmail.hzau.edu.cn (A.A.A.); xinyang@webmail.hzau.edu.cn (Y.X.); wangchunqun@webmail.hzau.edu.cn (C.W.); tingzhang1@webmail.hzau.edu.cn (T.Z.); ZCX19920102@webmail.hzau.edu.cn (C.Z.); yanxingrun@webmail.hzau.edu.cn (X.Y.); mubashar.hassan@webmail.hzau.edu.cn (M.H.); 2MOA Laboratory for Risk Assessment of Quality and Safety of Livestock and Poultry Products, Huazhong Agricultural University, Wuhan 430070, Hubei, China; omairshabbir105@gmail.com; 3Statistical Genomics Lab, College of Plant Science and Technology, Huazhong Agricultural University, Wuhan 430070, Hubei, China; ikramuaf35@outlook.com; 4Department of Parasitology, University of Veterinary and Animal Sciences, Lahore 54000, Pakistan; abdul.hafeez@uvas.edu.pk

**Keywords:** *Trichuris skrjabini*, mitochondrial genome, mitochondrial DNA, phylogenetic analysis

## Abstract

The complete mitochondrial (mt) genome of *Trichuris skrjabini* has been determined in the current study and subsequently compared with closely related species by phylogenetic analysis based on concatenated datasets of mt amino acid sequences. The whole mt genome of *T. skrjabini* is circular and 14,011 bp in length. It consists of a total of 37 genes including 13 protein coding genes (PCGs), two ribosomal RNA (rRNA) genes, 22 transfer RNA (tRNAs) genes, and two non-coding regions. The gene arrangement and contents were consistent with other members of the Trichuridae family including *Trichuris suis*, *Trichuris trichiura*, *Trichuris ovis,* and *Trichuris discolor*. Phylogenetic analysis based on concatenated datasets of amino acids of the 12 PCGs predicted the distinctiveness of *Trichuris skrjabini* as compared to other members of the Trichuridae family. Overall, our study supports the hypothesis that *T. skrjabini* is a distinct species. The provision of molecular data of whole mt genome of *T. skrjabini* delivers novel genetic markers for future studies of diagnostics, systematics, population genetics, and molecular epidemiology of *T. skrjabini.*

## 1. Introduction

Helminthiases, some of which are often referred as neglected tropical diseases, have accounted for devastating effects on human health. According to an estimate, helminths transmitted from soil (geo-helminths), most commonly whipworms (*Trichuris* spp.), hookworms (*Ancyclostoma* and *Necator* spp.), and roundworm (*Ascaris lumbricoides*), prevailing in less privileged regions of the world infect about one billion people [1]. Almost half billion people are infected with *Trichuris* spp., as a result about 0.64 million people suffer compromised life years [2]. Children with chronic infection develop clinical symptoms like dysentery, rectal prolapse, bloody diarrhea, and cognitive impairment and are more prone as compared to adults [3,4]. Whipworms are found to be in a wide range of hosts including bovines and caprines (*Trichuris ovis*, *Trichuris discolor*), dogs (*Trichuris vulpis*), pigs (*Trichuris suis*), humans (*Trichuris trichiura*), and non-human primates (*Trichuris* spp.) [5,6,7]. Transmission of these parasites is direct or occurs through a fecal–oral route. After ingestion, infective thick shelled embryonated eggs are hatched in the small intestine through gastric passage. Adult worms are developed (30–50 mm in length) from first stage larvae which are burrowed in the walls of caecum and proximal colon [8]. Whipworms are notorious for instigating significant economic losses and diseases, common in both humans and animals [9,10].

Currently on the basis of previous knowledge, morphological features of adults (pericloacal papillae and spicule), and the hosts harboring *Trichuris* species provides a foundation for their identification [11,12]. However, these criteria are not always able to reliably identify and differentiate the species [13,14]. *Trichuris skrjabini* was characterized and differentiated from *T. ovis* based on a morphological study and isoenzyme gel electrophoresis [15]. A molecular study based on Cytochrome *c* oxidase subunit I gene (*cox1*) and 16S rDNA of *T. skrjabini* showed homology with the *Trichinella* species [14]. However, rDNA sequences may not provide a suitable genetic marker to conduct systematics studies on nematodes due to enormous sequence polymorphism in these rDNA regions within the nematodes [16,17].

Mitochondrial (mt) DNA sequences provide a basis for genetic markers as well as studying systematics, population genetics, and evolutionary relationships due to maternal inheritance and rapid evolutionary change [18,19,20]. Delimitation due to high substitution rate and low population size of closely related species is another advantage of mtDNA sequences, leading to sorting of the speciation [21]. Therefore, comparative analysis can be useful to identify certain hidden species which cannot be differentiated by traditional methods [22]. Of mitochondrial genomes of nematodes to date, approximately 63 (complete or near-complete) have been reported and deposited in GenBank [23]. However, in the Trichuridae family complete mitochondrial genomes for only four species are available [24].

As yet, no mitochondrial genome has been reported for *T. skrjabini.* In the current study based on the previous investigations, we have reconstructed the phylogenetic relationships derived from known mt genomes of enoplid nematodes and also characterized the whole mitochondrial genome of *T. skrjabini* to determine the gene contents and their arrangements.

## 2. Materials and Methods

### 2.1. Parasite Collection and Isolation of DNA

Adult worms dwelling in the large intestine in sheep were collected from naturally infected sheep from a private farmer in Luotian, Hubei, PR China. The worms were extensively washed with physiological saline and identified initially as *Trichuris* spp. based on the morphological characteristics and predilection site [11,25]. The worms were then fixed in 70% ethyl alcohol and stored at −20 °C until further used for experiments. Standard sodium dodecyl/proteinase K treatment was done followed by mini column purification method (Wizard Genomic DNA Purification System, Promega, Beijing, China) to isolate the genomic DNA.

### 2.2. *Trichuris skrjabini* ITS-2 Amplification

As described previously by Anshay et al. [26], the ITS-2 region was amplified using NC5 and NC2 primers (Table 1) which was then identified by sequence analysis. A PCR reaction was carried out in a volume of 20 µL containing premix (Takara, Dalian, China), primers, and DNA template. The reaction conditions used were 94 °C initially for 5 min following the 35 cycles of 94 °C/30 s, 50 °C/30 s, and 72 °C/1 min, 72 °C/10 min final extension and reaction was stopped at 20 °C/5 min.

### 2.3. Amplification of Short and Long Fragments by PCR

The short fragments were amplified to obtain the partial sequences of *cox1*, *nad5*, and *cytb*. Primers used for amplification were based on mitochondrial genome sequences of *T. trichiura* and *T. suis* [24] and the primers were the same as reported in a previous study [27,28] (Table 1). 50 µL of total volume was used per amplicon in the PCR with reaction mixture containing 10 mM each of dNTP (Takara), 5 µL of 10× Thermopol reaction buffer (New England Biolabs, Ipswich, MA, USA), 1.25 U LATaq (Takara), each primer 2 µM (synthesized by TsingKe, Beijing, China), 34.75 µL dH₂O, and 2 µL of genomic DNA in a thermocycler (Biometra, Göttingen, Germany). The reaction was initiated with denaturation at 94 °C for 5 min followed by 35 cycles of denaturation at 94 °C, 50 °C annealing for 30 s, extension at 60 °C for 5 min, and final extension for 7 min at 60 °C. The amplified products were then sequenced directly (Sangon BioTech company, Shanghai, China) to obtain partial sequences.

Based on the partial sequences of *cox1*, *nad5,* and *cytb*, new primers were designed (Primer Premier 6.0) to amplify the long regions to obtain the complete mitochondrial genome (Table 1). They were amplified in three overlapping fragments ranging from *cox1* to *nad5*, *nad5* to *cytb* and *cytb* to *cox1*. The reaction mixture was the same as used before in PCR for amplifying short fragments. The conditions for amplification of long fragments were 92 °C denaturation for 5 min, then 35 cycles at 92 °C for 30 s, annealing at 51 °C for 30 s, extension at 65 °C for 7 min, followed by final extension at 65 °C for 7 min, and reaction was stopped at 16 °C for 1 min. The amplified products were then cloned into pGEM-TEASY vector (Promega, Madison, WI, USA) and were sequenced (Sangon BioTech company) by primer walking strategy. The entire mt genome sequence of *T. skrjabini* was obtained (GenBank accession number: MK333462).

### 2.4. Gene Annotation and Sequence Analysis

Manual assembly of the sequences obtained was performed by employing Clustal X 1.83 in contrast to *T. trichiura* complete mt genome sequence. The analysis of the open reading frames was executed by selecting invertebrate mitochondrial code using Open Reading Frame (ORF) Finder (http://www.ncbi.nlm.nih.gov/gorf/gorf.html) and subsequently comparison was done with mt genome sequences of other enoplids [24,29,30]. Genes were translated individually to get amino acid sequences by selecting invertebrate mitochondrial code in the software MEGA5 [31]. The alignment of the resulting amino acid sequences was performed against amino acid sequences of mt genomes of other nematodes utilizing Clustal X 1.83. For homologous genes, percentage of amino acid identity was also calculated on the basis of pairwise comparison. Codon usage was determined on the basis of relationships between nucleotide composition, codon families, and amino acid occurrence where the genetic codes are split into rich AT, GC, or neutral codons. Secondary structures of tRNA genes were identified to get rRNA genes by using the ARWEN tool (http://mbio-139 serv2.mbioekol.lu.se/ARWEN/) [32] and visual inspection [33].

### 2.5. Phylogenetic Analysis

The inferred amino acid sequences of 12 mt protein coding genes (PCGs) (other than *atp8* because of its absence in the outgroup) of *T. skrjabini* was aligned against mt amino acid sequences of other enoplid nematodes (*Trichinella spiralis*, GenBank accession number NC_002681; *T. trichiura*, GU385218; *T. suis*, GU070737; *T. ovis*, JQ996232; *T. discolor*, JQ996231; *Xiphinema americanum*, NC_005928; *Strelkovimermis spiculatus*, NC_008047; *Romanomermis iyengari*, NC_008693; *Romanomermis culicivorax*, NC_008640; *Romanomermis nielseni*, NC_008692; *Thaumamermis cosgrovei*, NC_008046; *Agamermis* sp., NC_008231; *Hexamermis agrotis*, NC_008828), where *Ascaris suum*, the chromadorean nematode (HQ704901) was selected as the outgroup [27]. The individual arrangement of groupings of amino acids got from mt protein coding sequences was performed utilizing the MAFFT 7.122 programming [34] and were fastened into a solitary dataset. Sequences which were adjusted vaguely were evaluated by a recently portrayed strategy [35]. Phylogenetic evaluation was steered by neighbor joining (NJ), maximum likelihood (ML), and maximum parsimony (MP) techniques as indicated by an earlier depicted strategy [36,37,38]. The program FigTree v. 1.4 (http://tree.bio.ed.ac.uk/programming/figtree) was utilized to develop the phylograms.

## 3. Results

### 3.1. ITS-2 Analysis

The acquired ITS-2 sequence of worm samples had a 99% match to a formerly published ITS-2 sequence of *T. skrjabini* (GenBank Accession no. AJ489248.1), identifying the worms gathered as *T. skrjabini*.

### 3.2. mt Genome Features

The entire mt genome sequence of *T. skrjabini* was 14,011 bp long (GenBank accession number: MK333462) (Figure 1). The mt genome of *T. skrjabini* was comprised of 13 PCGs (*cox1-3*, *nad1-6*, *nad4L*, *atp6,8* and *cytb*), 22 tRNA genes, two ribosomal RNA genes, and two non-coding sites (Table 2). The *atp8* gene is encoded (Figure 1), as is unvarying for enoplid nematodes [24,30]. The PCGs are translated in various directions, consistent with those depicted for *T. suis* and *T. spiralis* [24,30]. Apart from four PCGs (*nad2*, *nad5*, *nad4,* and *nad4L*) and six tRNA genes (*trnM, trnF, trnH, trnR, trnT, trnP*) encoded on the L-strand, every other gene is encoded on the H-strand. The AT-rich regions are sited among *nad1* and *trn*K, and *nad3* and *trnA*. The nucleotide synthesis of the whole mt genome was inclined toward A and T, with T being the most braced nucleotide and G being the slightest reinforced, which is steady with mt genomes of other enoplid nematodes for which mt genome information is accessible [24,30,39]. The content of A + T is 69.71% for *T. skrjabini* with a nucleotide composition of A: 34.75%, T: 34.96%, G: 13.88%, and C: 16.38%.

Moreover, the *T. skrjabini* mt genome has some overlaps among different tRNA genes and CDS regions (Table 2) fluctuating between 1–5 bp in smaller overlaps (*atp8–nad3; trnK–nad2; trnS_2_–trnN; trnF–trnM; trnG–atp8*), however some longer overlaps were also present, like *trnA* overlaps *trnS_2_* (43 bp), *trnF* overlaps *nad5* (56 bp), *trnD* overlaps *atp8* (57 bp), and *nad4L* overlaps *nad4* (104 bp).

### 3.3. Annotation

The *T. skrjabini* mt genome includes a total of 3582 amino acids exclusive of the termination codons. All of the PCGs possess complete termination codons. PCGs use ATG and ATA as start codons and utilize three termination codons (TAA, TGA, and TAG). Among the start codons, ATG has the highest frequency, being used by ten genes (*cox1, cox2, nad2, nad5, nad4, nad4L, nad6, cytb, atp6* and *cox3*), whereas *nad1*, *atp8*, and *nad3* genes use ATA as an initiation codon (Table 2). Terminal codon TAA is used by ten genes (*cox1, cox2, nad2, nad5, nad4, nad6, cytb, atp6, cox3* and *atp8*). *nad1* utilizes TGA as a termination codon and TAG is used as a termination codon by *nad4L* and *nad3* genes (Table 2).

The mt genome of *T. skrjabini* contains 22 tRNA genes ranging between 53 to 64 bp long. In comparison to other mt genomes of nematodes, tRNA genes are comparatively smaller to their corresponding genes which is due to DHU stem loop regions or condensed TV replacement loops [39]. *rrnS* is situated between *trnS*_1_ and *trnV*, whereas *rrnL* is located between *trnV* and *atp6*, the length for which is 566 bp and 840 bp respectively for *rrnS* and *rrnL.* AT contents for *rrnS* and *rrnL* genes were found to be 72.78 and 73.44%, respectively (Table 3). *T. skrjabini* mt genome also infers two non-coding regions represented as small non-coding region (SNCR) for smaller regions and longer non-coding region (LNCR) for longer regions, the AT contents for which are 70.50% and 81.81% respectively. SNCR is positioned between *nad3* and *trnA* whereas LNCR is situated between *nad1* and *trnK*. There is no clear evidence of authentic processing of these non-coding regions; however, they may play role in replication or transcription [40].

### 3.4. Phylogenetic Analysis

Based on the concatenated dataset of 12 PCGs amino acid sequences other than *atp8* (because of its absence in the outgroup), phylogenetic analysis was carried out for *T. skrjabini* (Figure 2) which generated similar topologies, irrespective of the approach used (NJ, MP, and ML). The bootstrap value for ML analysis was selected to be 100 whereas NJ and MP bootstrap values were 1000. The cut off value set for all the three methods was 95%. The uniform rates model was used for ML analysis and the number of differences model was used by NJ to infer the phylogenetic tree. The subtree–pruning–regrafting search method was used for MP analysis and the maximum trees to retain were 100. Within enoplid nematodes, two major clades were revealed (Trichuridae, Trichinellidae) and (Longidoridae, Mermithidae). *T. skrjabini* is more similar to *T. ovis* and *T. discolor* than *T. suis*, so it is closer to *T. ovis* and *T. discolor* in evolution. The results showed monophyly of the Trichuridae family (*T. suis*, *T. trichiura*, *T. ovis*, *T. discolor*) whereas the Trichinellidae family appeared to be less similar to *T. skrjabini* on the basis of evolution. Similarly, the results revealed the rejection of monophyly of other species belonging to Longidoridae and Mermithidae families in comparison to *T. skrjabini*.

## 4. Discussion

Whipworms cause parasitism of caecum in animals and humans in different parts of the world. Usually the identification of the species is carried out on the basis of their morphological and biometrical characteristics, and based on these characters more than 20 species of *Trichuris* have been reported [12]. Some of the pioneering studies of *Trichuris* species identification stated the fact that body length and body spicules were the most reliable characteristics for differentiation among the species [18,41]. Likewise, pericloacal papillae were considered to be useful criteria for species distinction by other researchers [42]. Though, still some of the species of this genus were challenging to identify based on these criteria [13].

In examination of taxonomic status, mt genome sequences deliver dependable genetic markers in comparative analyses [43,44,45,46,47,48]. Therefore, we determined the complete mt genome sequence of *T. skrjabini* in the current study. The size of the complete mt genome of *T. skrjabini* was 14,011 bp. The size of mt genomes of other species from Trichuridae family (*T. trichiura*, *T. suis*, *T. discolor*, *T. ovis*) ranged between 13,904 bp (the smallest for *T. ovis*) [37] and 14,436 bp (the largest for *T. suis*) [24]. Therefore, the size of the mt genome of *T. skrjabini* was found to be within the expected range. The gene contents (13 PCGs, two rRNAs, 22 tRNAs) and gene arrangement of *T. skrjabini* was also found to be consistent as in other studies, and the variation of the nucleotide sequence length among the species was also consistent [24,37] as the *Trichuris* spp. were isolated from different hosts.

On the basis of comparison between mt genomes of species of the Trichuridae family, the difference between the number of amino acid was almost negligible as *T. ovis* and *T. discolor* [37] possess 3577 and 3578 amino acids whereas the mt genomes of *T. trichiura* and *T. suis* possess 3559 and 3562 amino acids, respectively [24], and our study revealed the presence of 3582 amino acids. The usage of initiation and stop codons was also found to be consistent with some noticeable differences as *T. skrjabini* use only two start codons, i.e., ATG and ATA, where ATG, ATT, and ATA were utilized as start codons in the pig derived and human derived *Trichuris*. Moreover, *T. ovis* and *T. discolor* utilize four start codons, ATG, ATA, ATT, and TTG. In the case of termination codons, TAG and TAA were used as stop codons in all the four species of Trichuridae family, however, *T. skrjabini* possesses another stop codon TAG for the *nad1* gene, whereas other stop codons TAA and TAG were found to be consistent. These findings are consistent with previous studies, however, these marked differences are suggestive that *T. skrjabini* is distinct from other species of the Trichuridae family, hence providing new molecular data for the future comparative studies of mitochondrial genomes.

Phylogenetic study based on concatenated datasets of 12 PCGs for *T. skrjabini* by three different analysis methods (NJ, ML, and MP) also supported the evolution of the species and its distinctiveness as it is found to be more closer in relation to *T. ovis* and *T. discolor*. As reported by [37] these two species also may represent distinct species, however, some of studies provide a basis of differences among species which can represent different cryptic species originating from different hosts and geographical regions [25]. Thus, studies in the future may be able to explore variation in nucleotides in mtDNA and rDNA among populations from different regions and hosts.

## 5. Conclusions

The present study determined the complete mitochondrial genome of *T. skrjabini*. The molecular data presented in this study is distinct in the nucleotide sequences. Phylogenetic analysis also provides basis for distinctiveness of the species as compared to other species of the Trichuridae family which also supports our hypothesis. Therefore, the mtDNA data in the current study provides beneficial novel markers for population genetics and molecular epidemiological studies of *T. skrjabini*.

## Figures and Tables

**Figure 1 genes-10-00438-f001:**
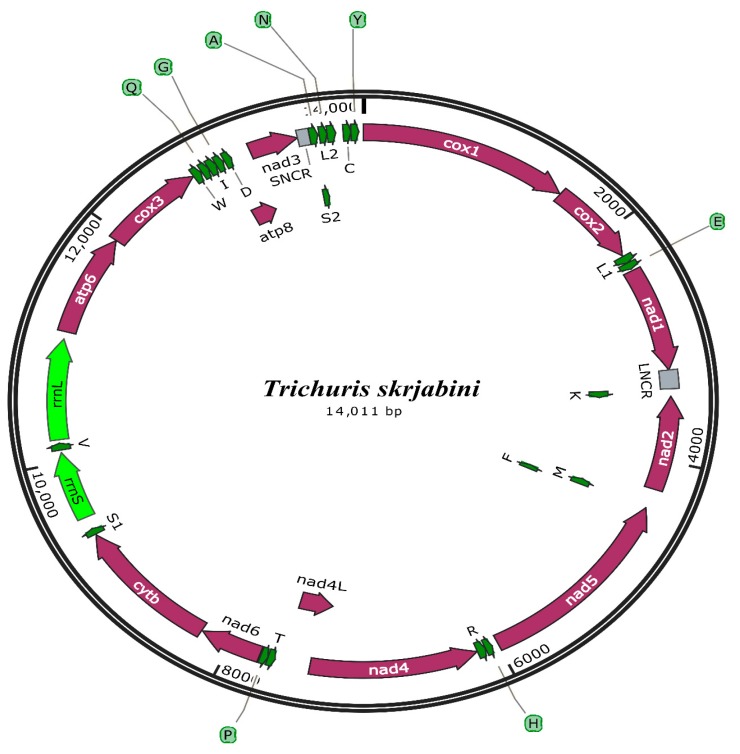
Mitochondrial genome structure of *T. skrjabini*. Genes represented as per standard nomenclature and tRNAs represented using one letter amino acid codes, with numerals differentiating each of the two leucine- and serine-specifying tRNAs (L1 and L2 for codon families UUR and CUA, respectively; S1 and S2 for codon families AGN and UGN, respectively). SNCR refers to small non-coding region and LNCR refers to longer non-coding region.

**Figure 2 genes-10-00438-f002:**
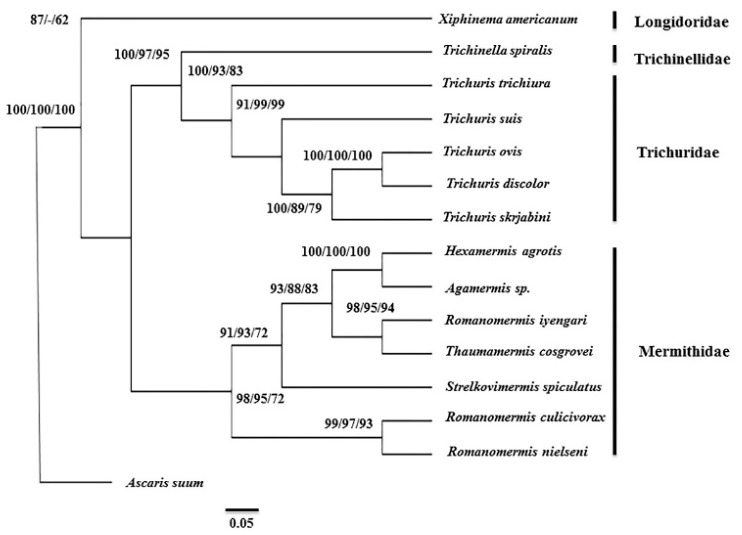
Phylogenetic tree inferred from concatenated amino acid sequence dataset of 12 protein coding genes (excluding *atp8* gene) among selected enoplid nematodes, using chromadorea nematode (*Ascaris suum*, NC_001327) as an outgroup. Phylogenetic relationships of *T. skrjabini* were inferred using the neighbor joining (NJ), maximum likelihood (ML), and maximum parsimony (MP) methods. The numbers along the branches indicate bootstrap values resulting from the analysis using NJ/ML/MP, where the values under 50 are given as “–”.

**Table 1 genes-10-00438-t001:** Primer sequences used for amplification of the ITS-2 region, short and long fragments of mitochondrial (mt) DNA of *Trichuris skrjabini*.

Primers	Sequence (5′ to 3′)
*ITS-2*	
NC5	GTAGGTGAACCTGCGGAAGGAT
NC2	TTAGTTTCTTTTCCTCCGCT
*Short PCR*	
*Trichuris_cox*1_F	GAAAGTGTTGGGGYAKAAAAGTTA
*Trichuris_cox*1_R	CAGGAAATCACAAAAAAATTGG
NAD-5F	CAAGGATTTTTTTGAGATCTTTTTC
NAD-5R	TAAACCGAATTGGAGATTTTTGTTT
Cytb-F	GAGTAATTTTTATAATACGAGAAGT
Cytb-R	AATTTTCAGGGTCTCTGCTTCAATA
*Long PCR*	
Cox1-F	CCTTGAAGCTTTGACACCTCC
Nad5-R	TGAGCTTTACGCAAGTTATGC
Nad5-F	CTAGCGTAAAAACTAGTGAGGTAAC
Cytb-R	GAGTGTGGGAACCAAGTGGA
Cytb-F	CTCTATGTGAAGCTTTTTGGGG
Cox1-R	TGGCAACTGCATGAGCAGATA

**Table 2 genes-10-00438-t002:** Structure of the mitochondrial genome of *T. skrjabini* and nucleotide positions of the starting and termination sites as well as the length of each gene and the number of encoded amino acids, starting and terminator codons of protein coding genes, and anticodons for tRNAs starting from *trnL_1_*.

Gene/Codons	Position and Sequence Length of Nt	Amino Acids	Start/Stop Codons	Anticodons
*cox1*	1–1545 (1545)	514	ATG/TAA	
*cox2*	1550–2233 (684)	227	ATG/TAA	
*trnL*_1_ (UUR)	2241–2302 (62)			TAA
*trnE*	2304–2360 (57)			TTC
*nad1*	2376–3245 (870)	289	ATA/TGA	
LNCR	3246–3401 (156)			
*trnK*	3402–3464 (63)			TTT
*nad2*	3463–4245 (783)	260	ATG/TAA	
*trnM*	4357–4419 (63)			CAT
*trnF*	4416–4471 (56)			GAA
*nad5*	4410–5996 (1587)	528	ATG/TAA	
*trnH*	6042–6095 (54)			GTG
*trnR*	6096–6159 (64)			TCG
*nad4*	6161–7396 (1236)	411	ATG/TAA	
*nad4L*	7293–7601 (309)	102	ATG/TAG	
*trnT*	7654–7708 (55)			TGT
*trnP*	7709–7762 (54)			TGG
*nad6*	7772–8230 (459)	152	ATG/TAA	
*cytb*	8239–9351 (1113)	370	ATG/TAA	
*trnS*_1_ (AGN)	9359–9411 (53)			GCT
*rrnS*	9515–10,080 (566)			
*trnV*	10,103–10,159 (57)			TAC
*rrnL*	10,189–11,028 (840)			
*atp6*	11,089–11,931 (843)	280	ATG/TAA	
*cox3*	11,933–12,706 (774)	257	ATG/TAA	
*trnW*	12,709–12,770 (62)			TCA
*trnQ*	12,778–12,834 (57)			TTG
*trnI*	12,836–12,898 (63)			GAT
*trnG*	12,899–12,955 (57)			TCC
*trnD*	12,966–13,022 (57)			GTC
*atp8*	12,951–13,169 (219)	72	ATA/TAA	
*nad3*	13,169–13,531 (363)	120	ATA/TAG	
SNCR	13,532–13,619 (88)			
*trnA*	13,620–13,680 (61)			TGC
*trnS_2_* (UGN)	13,638–13,692 (55)			TGA
*trnN*	13,691–13,747 (57)			GTT
*trnL*_2_ (CUA)	13,749–13,811 (63)			TAG
*trnC*	13,866–13,922 (57)			GCA
*trnY*	13,923–13,979 (57)			GTA

**Table 3 genes-10-00438-t003:** Composition of nucleotides and skew values of *T. skrjabini* mitochondrial protein coding genes.

Gene	A	G	C	T	A + T (%)	AT skew	GC skew
*cox1*	27.50	16.18	22.78	33.52	61.02	−0.09	0.16
*cox2*	34.06	13.74	23.09	29.09	63.12	0.07	0.25
*nad1*	28.96	13.21	20.45	37.35	66.31	−0.12	0,21
*nad2*	40.99	16.34	12.00	30.65	71.64	0.14	−0.15
*nad5*	41.14	14.36	13.35	31.12	72.26	0.13	−0.03
*nad4*	44.82	13.34	13.75	28.07	72.89	0.22	0.01
*nad4L*	42.71	10.67	17.79	28.80	71.51	0.19	0.25
*nad6*	23.09	12.85	13.28	50.76	73.85	−0.37	0.01
*cytb*	27.76	15.36	15.81	41.06	68.82	−0.19	0.01
*atp6*	33.33	18.62	18.62	38.19	71.52	−0.06	0.30
*cox3*	28.16	19.89	19.89	36.69	64.85	−0.13	0.13
*atp8*	30.59	16.43	16.43	42.92	73.51	−0.16	0.24
*nad3*	32.23	12.94	15.15	39.66	71.89	−0.10	0.07
*rrnS*	35.68	13.78	13.42	37.10	72.78	−0.01	−0.01
*rrnL*	38.33	12.73	13.80	35.11	73.44	0.04	0.04
LNCR	34.61	16.02	13.46	35.89	70.50	−0.01	−0.08
SNCR	47.72	14.54	13.63	34.09	81.81	0.16	0.50
Overall	34.75	13.88	16.38	34.96	69.71	−0.00	0.08

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
