# Peer review of "Characterization of the Complete Mitochondrial Genome of a Whipworm Trichuris skrjabini (Nematoda: Trichuridae)"

_genes, 2019, doi:10.3390/genes10060438_

Round 1

Reviewer 1 Report

It  is a laborious study on the mitochondria genome of T. skrjabini. The methods, results, and interpretation were excellent and logical. 

Author Response

Respected reviewer, Thanks a lot for your highly encouraging comments. The whole manuscript has been keenly revised. Your comments mean a lot to us. We are hopeful that our significant work will be a valuable addition for scientific community.  

Reviewer 2 Report

     The manuscript entitled as ‘Characterization of the complete mitochondrial genome of a whipworm Trichuris skrjabini (Nematoda: Trichuridae)’ by Ahmad et al. reports the whole mitochondrial genome of Trichuris skrjabini (Nematoda: Trichuridae). Although complete mitochondrial genomes of multiple whipworm species have been reported, this study has comparative genetic and phylogenetic importanc for nematodes and trichrids. The manuscript is well constructed and prepared. The current version, however, needs some revision to improve this significant report.

L17: ‘University of Veterinary and Animal Science’

L22-24: It is better to rewrite the initial two sentences. Unclear explanation on whipworms are presented.

L27: ‘It consists of …’

L31: ‘… based on concatenated datasets of …’

L40: ‘Helminthiases’ are not equal to ‘neglected tropical diseases’. For example, ‘Helminthiases, some of which are often referred as neglected tropical diseases, have accounted for ….’

L42: ‘…, and roundworm (Ascaris lumbricoides), prevailing …’

L43: ‘infect’ instead of ‘infects’.

L45: ‘Children with chronic infection …’ (?)

L50: ‘After ingestion, infective thick-shelled embryonated eggs are hatched in the small intestine …’ (?)

L52: ‘… from the first stage larvae…’  ‘Caecum’ is British English, and at other sentences, the authors used American English ‘cecum’.  Please use either English in a single manuscript.

L57: ‘species’ instead of ‘congeners’.

L59: Explanation of abbreviated genus names is not requested in the scientific description. Then, delete ‘(T. skrjabini)’ after the full species name. Instead, abbreviations for other words should be shown in the text independently from their exhibition in the Abstract section.

L60: ‘… based on Cytochrome c oxidase subnit I gene (cox1) and 16S rDNA of … Letters ‘c’ and ‘cox1’ in italic.

L61-63: Unclear sentence.

L66-67: Unclear sentence.

L69-72: Citation of references is needed. An example is’ International Helminth Genomes Consortium: Comparative genomics of the major parasitic worms. Nature Genet 51: 163-174; 2019’.

L79: The large intestine harbors adult worms, but adult worms could not harbor the large intestine.  ‘Adult worms dwelling in the large intestine’ (?)

P80: ‘physiological saline’ (?)

L81: Do the authors hypothesize mixed infection of multiple species? If not, please use ‘Trichuris sp.’ instead of ‘Trichuris spp.’

L84- : (Company name, city or province name, country name) for products. Sometimes, all information is available, and in another times only company name. Keep consistency.

L87: ‘… by Anshary et al. [25], ITS-2 ….’  It is easier to read the sentence.

L92: ‘… short and long fragments of mt DNA of Trichuris skrjabini.’

L95: ‘cox1’, ‘nad5’, and ‘cytb’ in italic, when they are gene names.

L97: Delete an excess space between ‘suis’ and ‘[23]’.

L97, L109, at least‘…, and the primers were the same as ….’ or ‘, and the primers were similar to those reported in the previous study’.

L103: For ‘˚C’, option key and ‘K’ key on the keyboard. Or choose from the list of special letter in the PC.

L137: ‘sp.’ in roman, not in italic.

L152: ‘identifying’ instead of ‘recommending’.

L171-172 [Figure 1]: Labelling ’14,000’ is unclear.

L189: ‘a terminal codon’

L237: ‘Trichuris spp.’ ‘spp’ in roman, not in italic.

L2353: ‘Liu et al. [36]’

L272-384: Check again all references. Keep consistency in format to describe publications.  Places of journal names, organism names in italic when the original titles followed this rule, etc. Sometimes inconsistent descriptions are seen.

By professional English edition, this manuscript could become a more valuable publication.

Author Response

L17: ‘University of Veterinary and Animal Science’

L22-24: It is better to rewrite the initial two sentences. Unclear explanation on whipworms are presented. Edited as per your suggestion.

L27: ‘It consists of …’

L31: ‘… based on concatenated datasets of …’

L40: ‘Helminthiases’ are not equal to ‘neglected tropical diseases’. For example, ‘Helminthiases, some of which are often referred as neglected tropical diseases, have accounted for ….’

L42: ‘…, and roundworm (Ascaris lumbricoides), prevailing …’

L43: ‘infect’ instead of ‘infects’.

L45: ‘Children with chronic infection …’ (?)

L50: ‘After ingestion, infective thick-shelled embryonated eggs are hatched in the small intestine …’ (?)

L52: ‘… from the first stage larvae…’  ‘Caecum’ is British English, and at other sentences, the authors used American English ‘cecum’.  Please use either English in a single manuscript.

L57: ‘species’ instead of ‘congeners’.

L59: Explanation of abbreviated genus names is not requested in the scientific description. Then, delete ‘(T. skrjabini)’ after the full species name. Instead, abbreviations for other words should be shown in the text independently from their exhibition in the Abstract section.

L60: ‘… based on Cytochrome c oxidase subnit I gene (cox1) and 16S rDNA of … Letters ‘c’ and ‘cox1’ in italic.

L61-63: Unclear sentence.

L66-67: Unclear sentence.

L69-72: Citation of references is needed. An example is’ International Helminth Genomes Consortium: Comparative genomics of the major parasitic worms. Nature Genet 51: 163-174; 2019’.

L79: The large intestine harbors adult worms, but adult worms could not harbor the large intestine.  ‘Adult worms dwelling in the large intestine’ (?)

P80: ‘physiological saline’ (?)

L81: Do the authors hypothesize mixed infection of multiple species? If not, please use ‘Trichuris sp.’ instead of ‘Trichuris spp.’

L84- : (Company name, city or province name, country name) for products. Sometimes, all information is available, and in another times only company name. Keep consistency.

L87: ‘… by Anshary et al. [25], ITS-2 ….’  It is easier to read the sentence.

L92: ‘… short and long fragments of mt DNA of Trichuris skrjabini.’

L95: ‘cox1’, ‘nad5’, and ‘cytb’ in italic, when they are gene names.

L97: Delete an excess space between ‘suis’ and ‘[23]’.

L97, L109, at least‘…, and the primers were the same as ….’ or ‘, and the primers were similar to those reported in the previous study’.

L103: For ‘˚C’, option key and ‘K’ key on the keyboard. Or choose from the list of special letter in the PC.

L137: ‘sp.’ in roman, not in italic.

L152: ‘identifying’ instead of ‘recommending’.

L171-172 [Figure 1]: Labelling ’14,000’ is unclear. The figure was designed based on the software, some of tRNA genes come over that label.

L189: ‘a terminal codon’

L237: ‘Trichuris spp.’ ‘spp’ in roman, not in italic.

L2353: ‘Liu et al. [36]’

L272-384: Check again all references. Keep consistency in format to describe publications.  Places of journal names, organism names in italic when the original titles followed this rule, etc. Sometimes inconsistent descriptions are seen. We are very thankful to you for your valuable suggestion. Corrections have been made.

By professional English edition, this manuscript could become a more valuable publication

Rep: Respected reviewer, Thanks a lot for your encouraging comments. The whole manuscript has been keenly revised. We have now addressed all the grammatical mistakes suggested by the reviewer. Further we also thoroughly studied the whole manuscript and revised accordingly in order to improve the language and grammar. We are quite hopeful that our presently revised manuscript will be considerable for your valuable comments.